# Unraveling the Prognostic Significance of BRCA1-Associated Protein 1 (BAP1) Expression in Advanced and Castrate-Resistant Prostate Cancer

**DOI:** 10.3390/biology14030315

**Published:** 2025-03-20

**Authors:** Norel Salut, Yaser Gamallat, Sima Seyedi, Joema Felipe Lima, Sunita Ghosh, Tarek A. Bismar

**Affiliations:** 1Department of Pathology and Laboratory Medicine, Cumming School of Medicine, University of Calgary, Calgary, AB T2N 4N1, Canada; norel.salut@ucalgary.ca (N.S.); yaser.gamallat@ucalgary.ca (Y.G.); sima.seyedi@ucalgary.ca (S.S.); joema.felipelima@ucalgary.ca (J.F.L.); 2Departments of Oncology, Biochemistry and Molecular Biology, Cumming School of Medicine, University of Calgary, Calgary, AB T2N 4N1, Canada; 3Arnie Charbonneau Cancer Institute, Cumming School of Medicine, University of Calgary, Calgary, AB T2N 4N1, Canada; 4Department of Medical Oncology, College of Health Sciences, University of Alberta, Edmonton, AB T6G 2R7, Canada; sunita.ghosh@ualberta.ca; 5Department of Public Health Sciences, Henry Ford Health, Detroit, MI 48202, USA; 6Tom Baker Cancer Center, Alberta Health Services, Calgary, AB T2N 4N1, Canada; 7Prostate Cancer Centre, Calgary, AB T2V 1P9, Canada; 8Department of Pathology, Alberta Precision Labs, Rockyview General Hospital, Calgary, AB T2V 1P9, Canada

**Keywords:** BAP1, prostate cancer, overall survival, cause-specific survival, ERG, PTEN

## Abstract

Prostate cancer remains one of the major cancers affecting men worldwide. Currently, reliable biomarkers to associate with disease progression are in need. We assessed the expression significance between BAP1 and prostate cancer patients’ outcomes. The results showed that lower expression of BAP1 is associated with worse overall survival and lethal disease vs. those with higher intensity. BAP1 expression was also correlated with other known genomic changes in prostate cancer. Incorporating BAP1 expression may have prognostic value in men affected by prostate cancer.

## 1. Introduction

Prostate cancer (PCa) remains a significant epidemiological concern and a leading cause of male mortality with millions of cases diagnosed worldwide annually [1,2]. A lack of early symptoms, timely screenings, and interventions leads to delayed detection, allowing the cancer to progress unchecked [3]. Despite advancements in health systems and treatment options in Western countries, aggressive forms of prostate cancer and lethality pose substantial challenges when diagnosed at later stages [4,5,6]. The search for key molecular players involved in its development and progression is crucial for improving diagnosis and treatment strategies are still needed [7].

Genomic alterations play a pivotal role in driving the aggressiveness and lethality of prostate cancer [8]. One of the most prevalent genomic alterations in PCa is the fusion of the ERG gene with TMPRSS2, promoting cancer development and progression [9]. Homozygous loss or the deletion of phosphatase and tensin homolog on chromosome 10 (PTEN) lead to the activation of pathways that drive tumor growth and invasion [10]. In conventional therapies like androgen deprivation therapy (ADT), resistance is often observed in the presence of amplifications or mutations in the androgen receptor (AR) [11]. Moreover, mutations in the P53 gene disrupt the regulation of cell growth and DNA repair mechanisms, fostering tumor progression [12]. These genomic alterations, either individually or in combination, contribute significantly to the aggressiveness and lethal potential of PCa and pose significant challenges in its clinical management and outcomes [13].

BRCA1-associated protein 1 (BAP1) is a tumor suppressor gene located on the short arm of chromosome 3 (3p21. 31–p21. 2) and encodes a deubiquitinase enzyme that plays a critical role in maintaining genomic stability and regulating cell cycle progression [14,15]. Germline mutations in BAP1 have been associated with an increased risk of multiple malignancies, including mesothelioma, renal cell carcinoma, and uveal and cutaneous melanoma [16].

In prostate cancer, we found contradictory data about the role of BAP1 expression in the disease’s pathogenesis and clinical outcomes. Several studies have reported that the presence of BAP1 mutations or the loss of expression in prostate cancer tissues leads to worse clinical outcomes, while the loss of BAP1 function has been shown to enhance AR activity, potentially contributing to the aggressive behavior of prostate cancer cells [17,18]. Others reported that high BAP1 expression is associated with prostate cancer development and progression [19]. Understanding the role of BAP1 in prostate cancer is not only important for unraveling the molecular mechanisms underlying the disease but also for identifying novel therapeutic targets. Targeting BAP1 or its associated signaling pathways may hold promise for developing more effective treatment strategies for prostate cancer patients.

The study aims to explore the involvement of BAP1 protein expression in prostate cancer lethality and uncover the molecular mechanisms behind its tumor suppressive effects.

## 2. Methods

### 2.1. Study Population, Tissue Microarray Construction, and Pathological Analysis

A cohort comprising 202 patients diagnosed with prostate cancer underwent transurethral resection of the prostate (TURP). Our patients were categorized into three groups. The incidental group comprised patients with no prior ADT therapy and Gleason grade group 1–3 (n = 61). Patients with a GG of more than 3 who underwent treatment post-TURP were categorized as the advanced group (n = 69). Those who received treatment before the TURP procedure and had advanced local disease with obstructive symptoms while on ADT were categorized as the castrate-resistant prostate cancer group (CRPC) (n = 72). The tissue samples were constructed into a tissue microarray (TMA). The histological diagnoses of individual cores on the TMA were verified by the study’s pathologist (TAB). BAP1 intensity expression was evaluated using a three-tiered system (1, weak; 2, moderate; 3, high intensity). Grade groups (GGs) were determined according to the 2018 WHO and ISUP grade group criteria by the study’s pathologist (TAB). Clinical data and outcomes were obtained from Alberta Cancer registries. This study was reviewed and approved by the University of Calgary Cumming School of Medicine Ethics Review Board in accordance with the 1964 Helsinki Declaration and its later amendments or comparable ethical standards (ethics Study REB Certification #HRE-BA.CC-16-0551).

### 2.2. Immunohistochemistry

Patients’ samples were constructed into tissue microarrays (TMAs) and 4 µm formalin-fixed paraffin-embedded (FFPE) sections were cut and mounted on glass slides, and then IHC was performed using a Dako Omnis auto-stainer (Agilent, Santa Clara, CA, USA) at the Anatomic Pathology Research Lab of Alberta Precision Research Labs (APRL) as a routine procedure. In brief, the slide was hydrated, and antigen retrieval was performed using Tris high epitope retrieval buffer (pH 9.0). Subsequently, slides were incubated with mouse monoclonal antibody Santa Cruze (sc-28383). The primary antibody was diluted at 1:100 dilution using Dako antibody diluent. Thirty-minute incubation was conducted for both primary and secondary antibody incubation. Mouse poly-linker (Agilent, USA) was applied to the sections after the primary antibody incubation. A DAB+ Substrate Chromogen system (Agilent, USA) was used as a post-incubation detection reagent. Hematoxylin was applied for counter-staining for nuclear staining. The slides were subjected to a series of ethanol dehydration and xylene treatments, followed by mounting with coverslips using Flo-TEXX mounting medium (Lerner Laboratories, Pittsburgh, PE, USA).

BAP1 was categorized into localization patterns, i.e., nuclear, cytoplasmic, and mixed localization, and further stratified based on weak/moderate and strong staining intensity.

Subsequently, PTEN, ERG, and p53 expressions were assessed using IHC. PTEN expression was characterized as either retained (normal expression) or lost (functional inactivation). ERG expression was categorized as either positive or negative, reflecting the presence or absence of ERG gene rearrangements. For p53, the IHC staining patterns were classified into wild-type expression (score 1) or aberrant expression (scores 0, 2, and 3). Furthermore, Gleason grades (GGs) were categorized into two groups: the low Gleason grade group (GS of 6–7) and the high Gleason grade group (GS of ≥8).

### 2.3. BAP1 Gene Expression Analysis in the Public Cancer Database

To explore the relative signature insight associated with *BAP1* gene expression and look for gene expression in PanCancer data using TNMplot [20], this data set directly compares tumor and normal samples by either grouping all specimens of the same category and running a Mann–Whitney U test or—in the case of the availability of paired normal and adjacent tumors—by running a paired Wilcoxon statistical test. Differential gene expression analysis was performed using the Biolake online tool [21] (https://biolake.ucalgary.ca/ accessed on 22 December 2023). Additional signature data were sourced from the Cancer Genome Atlas Prostate Adenocarcinoma (TCGA PRAD) database. Additionally, we investigated the relationship between BAP1 expression and top genomic alterations in TCGA PRAD data such as FXA1, IDH1, and SPOP, as well as gene fusions like ERG, ETV1, and ETV4 using the University of Alabama at Birmingham CANcer data analysis Portal (UALCAN) [22]. Moreover, BAP1 expression was correlated with prostate cancer (PCa) pathological stage, lymph node involvement, the number of affected lymph nodes, and residual tumor status using the UALCAN platform [22].

To enhance the analysis, Linkedomics [23] (https://www.linkedomics.org/ online tools accessed on 22 December 2023) were utilized to perform Gene Set Enrichment Analysis (GSEA), including the Gleason score and various pathological features associated with BAP1 expression.

### 2.4. Statistical Analysis

In this study, we employed various statistical methods to analyze the data. Descriptive statistics were utilized to summarize the characteristics of the dataset. For categorical data, we reported the frequency and proportions to provide an overview of the distribution. Meanwhile, for continuous data, we calculated and presented the mean and standard deviations, which offer insights into the central tendency and variability within the dataset.

To assess the significance of mean differences between two continuous measures, independent *t*-tests were used. Normality assumptions were tested, and parametric tests were conducted as per the central limit theorem. Welch’s *t*-test was used to determine the significant differences in expression levels between normal and primary tumors or tumor subgroups based on clinicopathological features. The fold changes between normal and tumor data were conducted using the Mann–Whitney test.

In the context of survival analysis, overall survival (OS) was defined as the duration from the time of diagnosis to the occurrence of death. Patients alive at the last date of follow-up were censored. Cause-specific survival (CSS) is a net survival measure representing cancer survival in the absence of other causes of death. For CSS, the event was considered if the patient died due to prostate cancer; all other patients were censored at the last date of follow-up or if they died due to other causes. Kaplan–Meier estimates and the corresponding 95% confidence intervals were reported for OS and CSS. Log-rank tests were used to compare the survival curves. Unadjusted and adjusted Cox regression analyses were conducted for OS and CSS outcomes. For the multivariable Cox’s regression model for OS and CSS, we adjusted for the Gleason score as one of the known confounding factors. The Gleason score was forced into the model irrespective of its statistical significance. BAP1 was considered as a single variable, and the combined effect of BAP1 with PTEN and ERG was analyzed. All these models were adjusted for the Gleason score. The hazard ratio (HR) and the corresponding 95% confidence intervals were reported. A *p*-value less than 0.05 was considered statistically significant. We applied two-sided *t*-tests to examine relationships and differences within the data. SPSS version 29 (IBM Corp. Released 2022. IBM SPSS Statistics for Windows, Version 29.0. Armonk, NY, USA: IBM Corp) was used to conduct all statistical analyses.

## 3. Results

### 3.1. Low BAP1 Expression Correlates with Poor Overall Survival (OS) and Cancer-Specific Survival (CSS)

We investigated of total of 202 PCa cases for BAP1 expression (Figure 1A,B).

In cases with a nuclear BAP1 pattern, a statistically significant correlation (*p* = 0.042) was observed between BAP1 intensity and PTEN expression. Strong BAP1 staining was associated with a higher incidence of PTEN loss (60%), whereas PTEN retention was more prevalent in the weak/moderate BAP1 staining group (62%). Similarly, the ERG variable demonstrated a significant difference (*p* = 0.030), with ERG positivity being more prevalent in cases with strong nuclear BAP1 staining (56%) compared to those with weak/moderate staining (33%). In contrast, p53 expression did not show a statistically significant correlation (*p* = 0.520) with either pattern of expression or intensity groups. Finally, Gleason grade exhibited a significant association (*p* = 0.012), with higher-grade Gleason groups more frequently displaying strong nuclear BAP1 staining (90.9%).

In the analysis of cytoplasmic and mixed BAP1 staining, we found no statistically significant relationships with PTEN, ERG, p53 expression, or Gleason grade across the different intensity groups. For cytoplasmic BAP1, PTEN loss was more frequent in the weak/moderate intensity group (64.3%) but was not observed in the strong intensity group (*p* = 0.082). Similarly, there were no notable differences in ERG positivity and Gleason grade distributions (*p* = 0.537 and *p* = 0.528, respectively).

Mixed BAP1 staining also showed no significant correlations, with PTEN loss seen in 41.2% of cases with strong intensity but completely absent in the weak/moderate group (*p* = 0.159). ERG positivity and Gleason grade were also not significantly different (*p* = 0.548 and *p* = 0.283), respectively.

In summary, nuclear BAP1 staining showed clear associations with PTEN loss, ERG positivity, and higher Gleason grades. However, no significant correlations were found for p53 expression or for cytoplasmic and mixed BAP1 staining. These findings underscore the importance of nuclear BAP1 staining in the characterization of prostate cancer.

For the remainder of the analysis, we focused on nuclear BAP1 expression since the initial data pointed to no significance for cytoplasmic expression.

Our results revealed that low BAP1 expression was significantly different between the three groups (*p* = 0.004) (Table 1). The mean intensity in the incidental group (n = 61) was 2.46 ± 0.535, advanced (n = 69) was 2.12 ± 0.676, and castrate-resistant (n = 72) was 2.17 ± 0.628. Interestingly, we found patients with high BAP1 expression (Figure 1C) who showed better survival outcomes in PCa (alive), but this was not statistically significant when comparing deceased and alive patient status with BAP1 expression categorized as low/high. Remarkably, 74.1% of low BAP1 expression cases exhibited advanced disease or were characterized as castration-resistant.

According to our outcome data and survival curves, BAP1 expression was categorized by scores into low risk (score 2, 3) and high risk (score 1) (Figure 1D,E). In the present cohort, reduced BAP1 intensity exhibited a significant association with unfavorable OS (HR 2.31, CI: 1.38–3.86, *p* = 0.001) and CSS (HR 2.44, CI: 1.24–4.78, *p* = 0.01) (Table 2).

### 3.2. Low Nuclear BAP1 Expression Combined with Either ERG Expression, PTEN Loss, P53, or AR Mutant Associated with Poor OS and CSS

Given the well-established associations of genomic alterations with the prognosis of prostate cancer patients, our objective was to explore the prognostic significance of BAP1 alone and in combination with common PCa genomic alterations such as PTEN, ERG, P53, and AR (Figure 2). Based on our results, the combination of low BAP1 expression and positive ERG expression (score 1) (Figure 2A) showed the most unfavorable OS (HR 3.79, CI: 1.62–6.82; *p* = 0.002) and CSS (HR: 4.72, CI: 1.65–13.52; *p* = 0.004) (Table 2).

Similarly, the combination of low BAP1 expression (score 1) and PTEN loss (score 0) was found to be associated with poor OS (HR: 2.83, CI: 1.35–5.91; *p* = 0.006) and CSS (HR: 4.30, CI: 1.64–11.28; *p* = 0.003). Figure 2B. Furthermore, using a multivariate analysis that adjusts for Gleason score, the combination of PTEN loss and low BAP1 expression was found to be significantly associated with inferior OS (HR: 2.34, CI: 1.07–5.12; *p* = 0.033) and CSS (HR 2.99, CI: 1.06–8.43, *p* = 0.038). Interestingly, improved prognosis and the highest survival rate were observed with high BAP1 expression (moderate and high) combined with no ERG expression or PTEN (moderate or high expression), P53 (score 0, 2), and AR (score 1, 2) (Figure 2).

### 3.3. PanCancer Data Analysis Revealed BAP1 Expression in Different Cancers

Exploring the PanCancer data, we found that *BAP1* gene expression was inconsistently expressed in 22 cancers (Figure 3A). Significant low expression was observed in adrenal, breast, esophagus, lung, ovary, prostate, renal, stomach, testicular, and thyroid cancers. However, a few tumors showed upregulation, such as acute myelogenous leukemia (AML), liver, pancreas, and renal cell carcinoma. Interestingly, this database shows that the BAP1 gene is significantly downregulated in cancer compared to non-adjacent normal tissues (Figure 3B).

### 3.4. BAP1 Gene Expression in TCGA PRAD

The gene expression analysis from TCGA PRAD data shows different trends compared to our patient cohort. The TCGA PRAD dataset analysis of BAP1 gene expression revealed an increase in BAP1 expression in the tumor samples when compared to adjacent or paired normal tissues (Figure 4A), moreover, the BAP1 gene upregulation was significantly correlated with increased Gleason score (Figure 4B) and common genomic aberrations such ERG fusion, ETV1 fusion, ETV4 fusion, FL1 fusion, FOXA1 mutation, IDH1 mutation, and SPOP mutation (Figure 4C). Furthermore, a similar trend was observed with TP53 mutant status compared to non-mutant or wildtype TP53 (Figure 4D). Also, a constant increase in *BAP1* gene expression was observed with nodal metastasis N0 and N1 (Figure 4E).

### 3.5. Gene Set Enrichment Analysis of BAP1 Gene Revealed Potential Role in PCa

Differential gene expression analysis was performed (Figure 4F) for BAP1 using the TCGA PRAD dataset. Differential gene expression analysis revealed potential tumorigenesis-associated candidates as presented in the top BAP1 gene expressions shown with a heatmap in Figure 5A,B. Furthermore, the gene set enrichment analysis was further performed to identify the molecular function as shown in Figure 6A,B. Our data revealed that BAP1 gene expression is negatively correlated with many potential pathways involved in tumor progression, such as the WNT pathway, angiogenesis, P53 pathway, and TGF beta signaling.

## 4. Discussion

Our findings document that BAP1 nuclear expression in PCa has significant prognostic value. Other patterns of expression (cytoplasmic and mixed) show no correlation with clinical outcome or biomarker associations. Our cohort showed that weak BAP1 nuclear expression is associated with poorer prognosis, higher Gleason scores, and advanced tumor stages. This suggests the potential for BAP1 as an adverse prognostic factor. In contrast, strong BAP1 expressions indicate a favorable clinical outcome.

Previous studies show that *ERG* gain, TMPRSS2 fusion, and *PTEN* loss are associated with poor clinical outcomes in PCa patients [23,24]. Our data also show that weak BAP1 nuclear expression, alone and when combined with ERG rearrangement or PTEN loss, correlates with poor clinical outcomes.

BAP1 functions as a deubiquitinating enzyme, stabilizing target proteins and preventing their degradation. This mechanism is crucial in maintaining the tumor suppressive function of proteins like PTEN. BAP1 regulates PTEN stability by removing ubiquitin, preventing its degradation, and thereby inhibiting the PI3K/AKT signaling pathway. When BAP1 expression is lost or reduced, PTEN becomes increasingly ubiquitinated and degraded, leading to unchecked cell proliferation and tumor progression by PI3K/AKT pathway activation.

BAP1 interacts indirectly with ERG, a key oncogene in prostate cancer. ERG overexpression, often driven by the TMPRSS2-ERG gene fusion, is exacerbated by low BAP1. BAP1 normally counteracts the activity of Polycomb Repressive Complex 1 (PRC1) by deubiquitinating histone H2A at lysine 119 (H2AK119ub), a marker of gene silencing. Low BAP1 leads to the accumulation of H2A ubiquitination, reinforcing the PRC1-mediated repression of tumor suppressor genes, thus creating a permissive environment for ERG overexpression and tumor progression.

Our study suggests that BAP1 is independently associated with adverse PCa outcomes, indicating a broader influence on gene expression, cell cycle regulation, and epigenetic changes. Low BAP1 expression promotes genomic instability by failing to regulate DNA repair pathways or by altering the expression of other tumor suppressor genes, contributing to an aggressive cancer phenotype. Additionally, it may create a permissive epigenetic environment that fosters the activation of additional oncogenic signaling pathways, driving tumor progression. BAP1’s ability to maintain genomic integrity and regulate protein stability strongly supports its role as a tumor suppressor and highlights its important role in PCa biology.

Additionally, our data also show BAP1 as a risk factor regardless of p53 and AR mutation status. However, a favorable outcome was observed with combined BAP1 and low-risk p53 or AR status.

Several studies that highlight the significance of BAP1 expression in prostate cancer [25,26] show conflicting findings. While most authors believe that BAP1 acts as a tumor suppressor gene, others consider it as an oncogene [14,27,28,29]. Recent findings show that BAP1 suppresses prostate cancer progression by deubiquitinating and stabilizing PTEN, inhibiting the PI3K-AKT-mTOR pathway and suppressing trophoblastic EMT [14]. Our data support this view regarding BAP1 as a tumor suppressor.

Genomic alterations, such as ERG rearrangements, PTEN loss, p53 mutations, and AR aberrations, have a significant impact on the prognosis of PCa. ERG rearrangements often signal more aggressive disease behavior, while PTEN loss correlates with heightened tumor aggressiveness and resistance to therapies [30]. TP53 mutations lead to higher-grade tumors and increased genomic instability, contributing to disease progression and poorer outcomes [31]. Concurrently, androgen receptor mutations or amplifications are linked to treatment resistance and the development of castration-resistant prostate cancer (CRPC), intensifying disease severity [11,32]. Our data did not all point in this direction, reflecting sample size issues. This suggests that such models may have clinical applicability in some clinical scenarios but likely need to be confirmed in subsequent studies, as more combinations or expansions of the sample data may help create a more comprehensive tissue-based prognostic model.

A previous study reported findings that contradict our data [17]. Considering the substantial difference in sample size and the biological significance of our findings, it is important to highlight that our analysis was conducted on a different cohort in terms of the demographics and outcomes assessed. In the cohort referenced, patients were treated surgically (radical prostatectomy) with biochemical recurrence as the assessed outcome. Conversely, our cohort is composed of non-surgically treated patients with assessed outcomes related to OS and CCS.

It is well known, for example, that ERG is not prognostic in most surgical cohorts, but has significant prognostic value in non-surgical and hormonally treated cohorts [33,34]. We believe this distinction is crucial, as it may account for some of the differences observed between our study and the earlier publication referenced.

The TCGA PRAD data show higher BAP1 mRNA levels in PCa samples (combined), which is inconsistent with our results. However, PanCancer supports our finding of lower BAP1 expression in benign versus cancer tissues and tissue-based expression data of weak BAP1 expression linked with worse clinical outcomes. This adds more value to our tissue-based signatures. The difference may be related to TCGA assessing mRNA versus protein expression, as in this study. Additionally, numerous factors may contribute to the discrepancy between mRNA abundance and actual protein expression. Post-transcriptional modifications, including RNA processing and stability, influence the fidelity of mRNA representation [35].

## 5. Conclusions

Our data suggests that integrating BAP1 with PTEN deletion and ERG rearrangements significantly enhances the accuracy of prognostic models. However, adding more genomic factors may allow better risk stratification and prediction of disease progression in prostate cancer patients.

## Figures and Tables

**Figure 1 biology-14-00315-f001:**
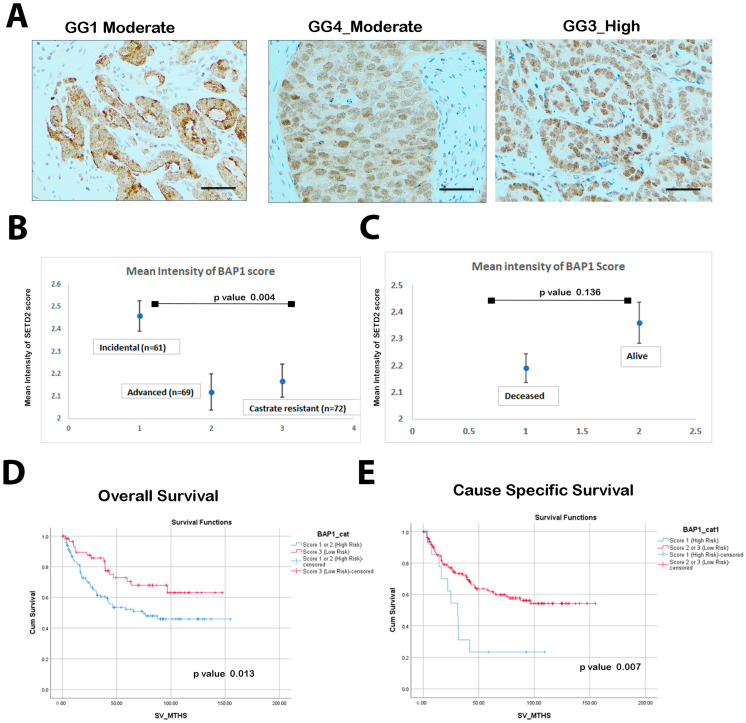
Lower Nuclear BAP1 expression is associated with poor overall and cause-specific survival. (**A**) Immunohistochemistry assessment represents BAP1 nuclear staining in prostate tissues with high and low expression patterns. Top left (GG1_High) panel showing Acinar Adenocarcinoma (AA) Gleason score 3 + 3 (GG1) with moderate cytoplasmic and membranous staining patterns. In the middle (GG4_Moderate), AA Gleason score 4 + 4 (GG4) displaying a moderate intensity of nuclear staining and weak cytoplasmic expression. The right top panel IHC image (GG3_High) represents a strong nuclear staining pattern with AA Gleason score 4 + 3 (GG3). (Scale bar = 100 μm). (**B**) Boxplot shows the expression intensity of BAP1 (mean ± standard error, SE) in incidental, advanced, and castrate-resistant PCa tissues. BAP1 protein expression levels were scored through IHC. Each sample was scored semi-quantitatively using a three-tiered system (weak—1; moderate—2; strong—3). The error bars indicate the standard deviation of the mean. (**C**) Boxplot representing the BAP1 intensity (mean ± SE) comparing patients to survival status (deceased and alive). (*p*-value = 0.136) (**D**) Kaplan–Meier curve (KM) representing the BAP1 expression and associated overall survival (OS) (**E**) KM curves representing the BAP1 expression and associated cause-specific survival (CSS).

**Figure 2 biology-14-00315-f002:**
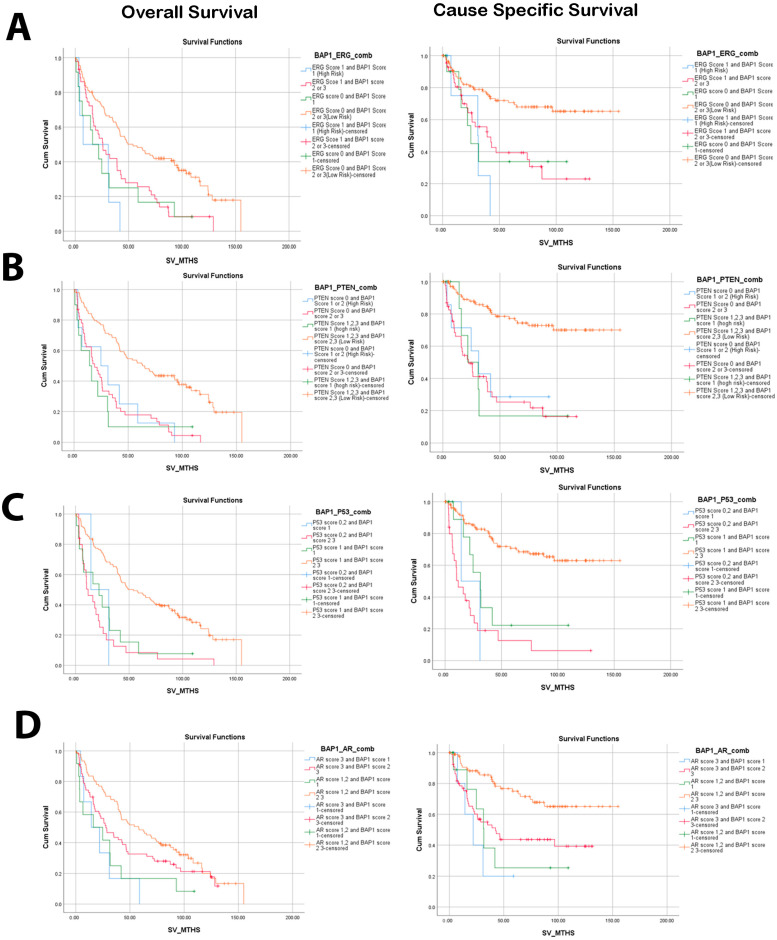
Overall (OS) and cause-specific survival (CSS) outcomes associated with BAP1 expression and major genomic alterations in our cohort. Kaplan–Meier curve showing the overall survival and cause-specific survival of PCa patients in relation to BAP1 protein expression levels combined with ERG (**A**), PTEN (**B**), P53 (**C**), and AR (**D**). BAP1 score 1 (weak) presented as high risk, while expression scores 2 and 3 (moderate and high) were low risk.

**Figure 3 biology-14-00315-f003:**
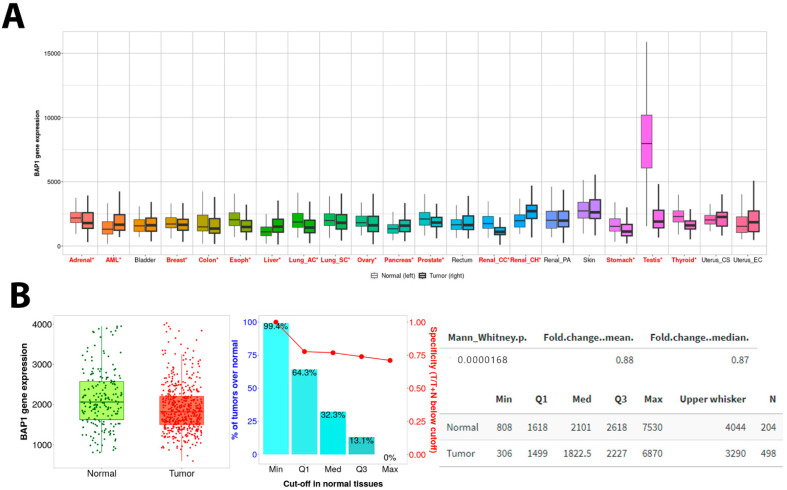
BAP1 gene expression in PanCancer data. (**A**) Boxplots showing PanCancer data analysis of 22 cancer types. TNM Plot PanCancer database includes 56,090 unique samples from GEO, GTex, TCGA, and TARGET databases. This includes normal (n = 15,648) and tumor (n = 40,442) samples. Data were analyzed by Mann–Whitney U test. * Significant differences are marked with red. (**B**) Boxplot showing the BAP1 gene expression between normal prostate tissues from non-cancerous patients (n = 204) and prostate tumors (n = 498) using TCGA PRAD data. Mann–Whitney U test, *p* value 0.0000168. Acute Myeloid Leukemia (AML), Lung adenocarcinoma AC, Lung squamous cell carcinoma (SC), Renal papillary cell carcinoma (Renal PA), Kidney chromophobe (Renal CH), Renal Clear Cell Carcinoma (Renal CC), Uterus CS (uterine carcinosarcoma) and Uterus EC (Endometrial carcinoma).

**Figure 4 biology-14-00315-f004:**
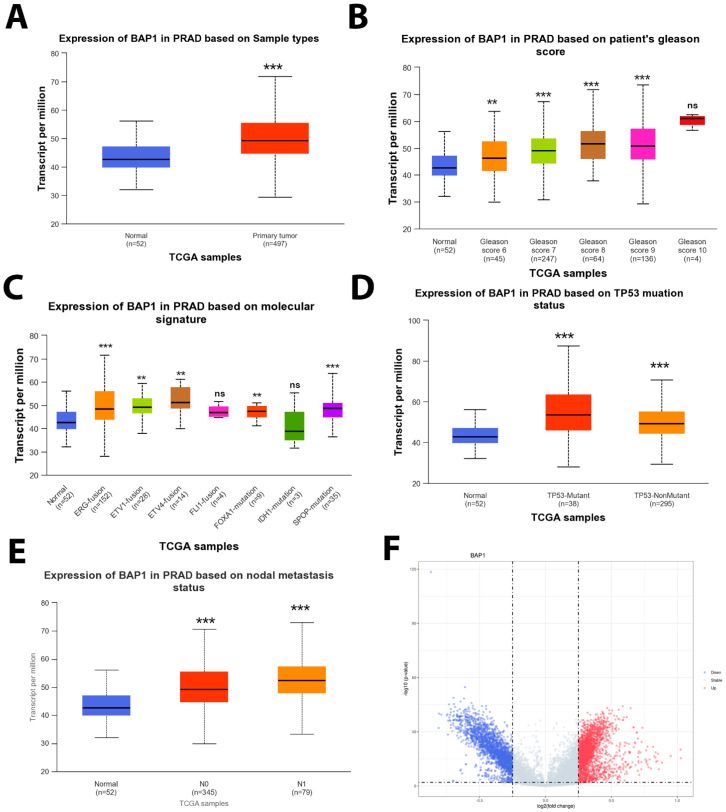
BAP1 mRNA expression in TCGA PRAD dataset. (**A**) Boxplot showing the mRNA expression levels of *BAP1* in normal (paired/adjacent tissue) vs. tumor tissue. (**B**) BAP1 expression according to Gleason score. (**C**) Boxplot showing the Bap1 expression and common genomic alterations in PCa such as ERG fusion (n = 152), ETV1 fusion (n = 28), ETV4 fusion (n = 14), FL1 fusion (n = 4), FOXA1 mutation (n = 9), IDH1 mutation (n = 3), and SPOP mutation (n = 35). (**D**) Boxplot analysis of BAP1 expression with TP53 status in TCGA PRAD dataset. (**E**) Boxplot showing the BAP1 mRNA expression levels compared to nodal metastasis status: N0—no regional lymph node metastasis; N1—metastases in 1 to 3 axillary lymph nodes; N2—metastases in 4 to 9 axillary lymph nodes; N3—metastases in 10 or more axillary lymph nodes. All BAP1 gene expression was analyzed using transcript per million (TPM). Welch’s T-test was used to determine the significant differences in expression levels between normal and primary tumors or tumor subgroups based on clinicopathological features. Asterisks indicate *p*-values: *p* < 0.01 **. *p* < 0.001 ***, ns not significant. (**F**) Volcano plot showing fold change genes associated with BAP1 gene expression.

**Figure 5 biology-14-00315-f005:**
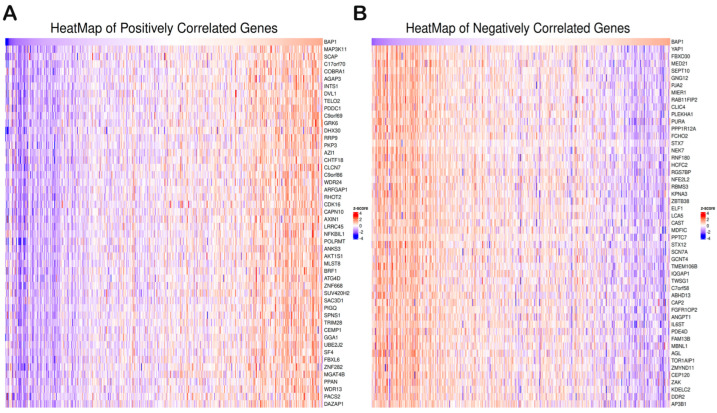
Differential gene expression analysis of *BAP1* in TCGA-PRAD. (**A**) Heat map of the top 50 positively correlated genes associated with BAP1. (**B**) Heat map showing the top 50 negatively correlated genes.

**Figure 6 biology-14-00315-f006:**
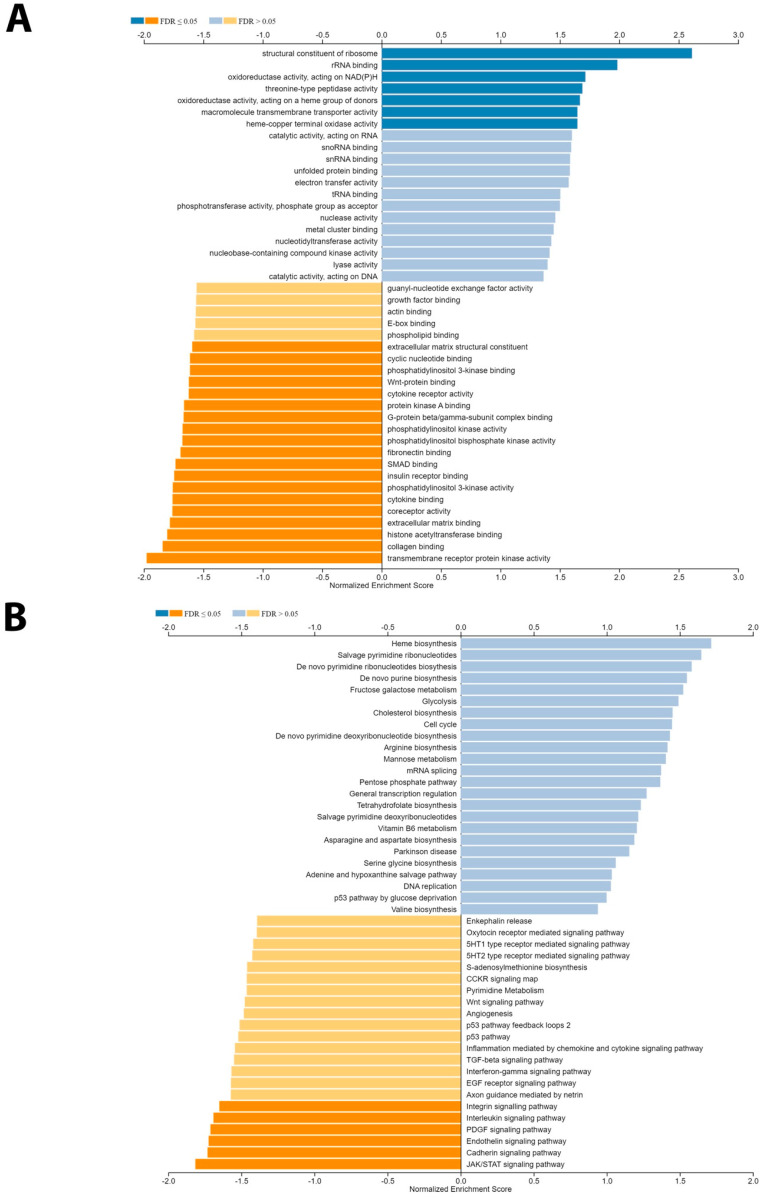
Gene set enrichment analysis. (**A**) Bar plots showing the gene set enrichment analysis grouping of genes that are positively correlated with the high expression of BAP1. (**B**) Boxplot showing the enrichment analysis using pathway_Panther data analysis for gene function analysis.

**Table 1 biology-14-00315-t001:** Demographic and distribution data for the prostate cancer patients assessed for BAP1 expression.

Variables	Score 1 (n = 22)	Score 2 (n = 110)	Score 3 (n = 70)
Survival			
Deceased	17 (85.0)	78 (71.6)	44 (64.7)
Alive	13 (46.4)	31 (32.6)	13 (18.3)
Cancer subgroup			
Incidental	1 (4.5)	31 (28.2)	29 (41.4)
Advanced	12 (54.5)	37 (33.6)	20 (28.6)
Castrate-resistant	9 (40.9)	42 (38.2)	21 (30.0)
PTEN			
Loss	9 (45.0)	30 (29.4)	16 (24.6)
Gain	11 (55.0)	72 (70.6)	49 (75.4)
ERG			
Negative	13 (65.0)	74 (71.80	50 (76.9)
Positive	7 (35.0)	29 (28.2)	15 (23.1)
AR			
Score 1, 2	14 (63.6)	54 (50.5)	50 (74.6)
Score 3	8 (36.4)	53 (49.5)	17 (25.4)
P53			
Score 1	14 (73.7)	88 (82.2)	60 (89.6)
Score 0, 2	5 (26.3)	19 (17.8)	7 (10.4)

BAP1 score: 1—weak; 2—moderate; and 3—high. PTEN score: 0—negative (loss); 1, 2, 3—positive. ERG-risk groups: positive—gain (high risk); negative—loss (low risk). p53 score 0 = normal; P53 scores 0, 2, 3 = mutant. AR scores 1, 2 = high risk; AR score 3 = low risk.

**Table 2 biology-14-00315-t002:** The association between BAP1 expression and overall (OS) and cause-specific survival (CSS) with Gleason score, ERG, and PTEN expression.

Variables	Overall Survival HR (95% CI)	*p*-Value	Cause-Specific Survival HR (95% CI)	*p*-Value
PTEN (score 1, 2 or 3)				
Loss—score 0	2.71 (1.89–3.89)	<0.0001	4.50 (2.76–7.34)	<0.0001
ERG (negative)				
Positive	1.93 (1.34–2.77)	<0.0001	2.53 (1.55–4.11)	<0.0001
GS (≤6)				
GS 3 + 4	2.03 (0.98–4.20)	0.056	24.18 (2.70–216.62)	<0.0001
GS 4 + 3	1.41 (0.72–2.77)	0.322	9.68 (1.01–93.15)	0.049
GS 8	5.09 (2.47–10.49)	<0.0001	71.43 (8.68–588.10)	<0.0001
GS 9, 10	6.02 (3.80–9.51)	<0.0001	107.23 (14.73–780.83)	<0.0001
BAP1 low-risk (score 2, 3)				
BAP1 high risk (score 1)	2.31 (1.38–3.86)	0.001	2.44 (1.24–4.78)	0.01
Combination PTEN and BAP1 (PTEN score 1, 2, 3 and BAP1 score 2, 3)				
PTEN score 0 and BAP1 score 1	2.83 (1.35–5.91)	0.006	4.30 (1.64–11.28)	0.003
PTEN score 0 and BAP1 score 2, 3	3.03 (2.05–4.46)	<0.0001	5.43 (3.19–9.24)	<0.0001
PTEN score 1–3 and BAP1 score 1	3.57 (1.77–7.20)	<0.0001	4.95 (1.88–13.03)	0.001
Combination PTEN and BAP1 (PTEN score 1, 2, 3 and BAP1 score 2, 3) *				
PTEN score 0 and BAP1 score 1	2.34 (1.07–5.12)	0.033	2.99 (1.06–8.43)	0.038
PTEN score 0 and BAP1 score 2, 3	1.63 (1.06–2.53)	0.027	1.91 (1.10–3.33)	0.023
PTEN score 1–3 and BAP score 1	1.36 (0.65–2.86)	0.421	1.13 (0.42–3.03)	0.808
Combination ERG and BAP1 (ERG negative and BAP1 score 0–2)				
ERG positive and BAP1 score 1	3.79 (1.62–6.82)	0.002	4.72 (1.65–13.52)	0.004
ERG positive and BAP1 score 2, 3	1.97 (1.34–2.90)	0.001	2.66 (1.57–4.50)	<0.0001
ERG negative and BAP1 score 1	2.43 (1.29–4.60)	0.006	2.83 (1.18–6.78)	0.02-
Combination ERG and BAP1 (ERG negative and BAP1 score 0–2) *				
ERG positive and BAP1 score 1	1.65 (0.69–3.95)	0.257	1.41 (0.49–4.06)	0.53
ERG positive and BAP1 score 2, 3	1.20 (0.79–1.84)	0.395	1.12 (0.65–1.93)	0.69
ERG negative and BAP1 score 1	1.37 (0.71–2.64)	0.347	1.12 (0.46–2.72)	0.799

* Adjusted for Gleason grade group. BAP1 score: 1—weak; 2—moderate; and 3—high. PTEN score: 0—negative (loss); score 1–3—weak to high. ERG negative = no expression; positive = expressed.

## Data Availability

The data can be shared upon request.

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
