# Peer review of "Unraveling the Prognostic Significance of BRCA1-Associated Protein 1 (BAP1) Expression in Advanced and Castrate-Resistant Prostate Cancer"

_biology, 2025, doi:10.3390/biology14030315_

Round 1
Reviewer 1 Report
Comments and Suggestions for Authors
Dear Authors,
The present manuscript has an in-depth and insightful investigation into the prognostic significance of BRCA1-associated protein 1 (BAP1) expression in advanced and castration-resistant prostate cancer. The study included a robust cohort of prostate cancer samples across various stages, encompassing both advanced and castration-resistant types. The authors discovered a noteworthy inverse relationship between BAP1 nuclear expression and the aggressiveness of the tumors, as well as with poorer clinical outcomes. Specifically, they found that lower levels of BAP1 nuclear expression were associated with more aggressive tumor characteristics, while higher levels of BAP1 expression correlated with less aggressive forms of prostate cancer and better patient prognosis.
In addition to this, the study revealed that lower BAP1 expression was linked to unfavorable overall survival and cause-specific survival rates. Conversely, higher BAP1 nuclear expression was associated with improved survival outcomes, further supporting the hypothesis that BAP1 may function as a tumor suppressor in prostate cancer. These findings are significant, as they point to BAP1’s potential role in the progression of prostate cancer and its value as a prognostic marker.
Despite the availability of various biomarkers for prostate cancer, the need for reliable molecular markers that can accurately stratify risk and predict disease progression is becoming increasingly critical. This study offers compelling evidence that BAP1 could serve as an important molecular biomarker for refining prognostic models. By incorporating BAP1 expression into risk stratification, clinicians may be able to better identify patients at higher risk for aggressive disease and guide more targeted treatment strategies. Ultimately, the study provides a promising avenue for improving the management of prostate cancer, suggesting that BAP1 could play a key role in developing more effective approaches to identifying and treating the disease.
Minor Comment: Please check for minor spelling edits for ex: line 43 has CCS instead of CSS.
Thank you
Author Response
Dear Biology Editor:
Please find our detailed response to reviewer comments and revised manuscript as suggested. “Unraveling the Prognostic Significance of BRCA1-associated protein 1 (BAP1) Expression in advanced and castrate resistance Prostate Cancer “Biology 3469449R1.
We hope you will find our responses adequate for the manuscript to be accepted in biology.
All authors have approved this revised version.
Sincerely.
Tarek A. Bismar, MD
Professor, University of Calgary
Detailed responses:
The present manuscript has an in-depth and insightful investigation into the prognostic significance of BRCA1-associated protein 1 (BAP1) expression in advanced and castration-resistant prostate cancer. The study included a robust cohort of prostate cancer samples across various stages, encompassing both advanced and castration-resistant types. The authors discovered a noteworthy inverse relationship between BAP1 nuclear expression and the aggressiveness of the tumors, as well as with poorer clinical outcomes. Specifically, they found that lower levels of BAP1 nuclear expression were associated with more aggressive tumor characteristics, while higher levels of BAP1 expression correlated with less aggressive forms of prostate cancer and better patient prognosis.
In addition to this, the study revealed that lower BAP1 expression was linked to unfavorable overall survival and cause-specific survival rates. Conversely, higher BAP1 nuclear expression was associated with improved survival outcomes, further supporting the hypothesis that BAP1 may function as a tumor suppressor in prostate cancer. These findings are significant, as they point to BAP1’s potential role in the progression of prostate cancer and its value as a prognostic marker.
Despite the availability of various biomarkers for prostate cancer, the need for reliable molecular markers that can accurately stratify risk and predict disease progression is becoming increasingly critical. This study offers compelling evidence that BAP1 could serve as an important molecular biomarker for refining prognostic models. By incorporating BAP1 expression into risk stratification, clinicians may be able to better identify patients at higher risk for aggressive disease and guide more targeted treatment strategies. Ultimately, the study provides a promising avenue for improving the management of prostate cancer, suggesting that BAP1 could play a key role in developing more effective approaches to identifying and treating the disease.
Minor Comment: Please check for minor spelling edits for ex: line 43 has CCS instead of CSS.
Response: We re-checked spelling and edited the manuscript accordingly.
Section 2.1 is a bit unclear, which makes it harder to follow.
The use of two-tailed t-tests raises some concerns about statistical assumptions. It’s not clear whether the authors checked for normality before applying the test, which is important because violations of this assumption could affect the results. If the groups had unequal variances, Welch’s t-test would have been a better choice. Since multiple t-tests seem to have been used, it would be helpful to clarify whether any correction for multiple comparisons (such as a Bonferroni adjustment) was applied. If the data are not normally distributed, a nonparametric test like the Mann-Whitney U test might be more appropriate. Also, if other factors influence the comparisons, ANCOVA or regression models could provide a more reliable approach.
Response:
Normality assumptions were checked, and the data were skewed, however, we used parametric tests based on the central limit theorem (“When large samples usually greater than thirty are taken into consideration then the distribution of sample arithmetic mean approaches the normal distribution irrespective of the fact that random variables were originally distributed normally or not.”). The Bonferroni adjustments were made. For comparing the fold change, we used Mann-Whitney change. We have made the changes in the method section.
Figures 5C and 5D are too small to read—perhaps they could be rearranged for better visibility?
Response: The figure was spitted for better readability as per the reviewer suggestion.
. The discussion could be clearer in some areas. Many sentences are long and complicated, making it difficult to follow the key points. Breaking them into shorter sentences would improve readability. Some ideas are also repeated multiple times (for example, references to BAP1 as a prognostic factor and its connection to PTEN/ERG), which could be streamlined.
Response: The discussion section was re-edited and improved.
The authors mention that sample size is a limitation but don’t quantify how it impacts statistical power. Discussing effect sizes and confidence intervals would help strengthen their argument. They also note differences between their patient cohort and others but don’t fully explore potential confounding factors beyond treatment differences.
Response:
Sample size for the Bap1 population is 202 with 22 subjects having a score of 1 compared to 180 for score 2,3. When we ran the survival analysis the hazard ratio which we obtained had a tighter confidence interval but when combined with other markers we did see statistical significance but had wider confidence intervals, however, the statistical significance was maintained, indicating that the results were significant even with small sample size. The result for this study needs to be confirmed using independent dataset with larger sample size to establish association. Our results cannot be generalized as one of the limitations of the smaller sample size.
We adjusted for one of the major confounding factors: Gleason score in our multivariate analysis.
The discussion also touches on inconsistencies with a larger cohort study (~17,000 patients), but the explanation given (differences in treatment and cohort characteristics) feels a bit vague. It would be useful to consider whether other factors, such as patient selection criteria or analytical methods, could explain these discrepancies. Additionally, while the authors state that their findings align with Pan-Cancer data, they don’t clarify whether this applies specifically to prostate cancer or includes other tumor types, which could lead to confusion.
Response:
In the cohort referenced, patients were treated surgically (radical prostatectomy) with biochemical recurrence as the assessed outcome. Conversely, our cohort is composed of non-surgically treated patients with assessed outcomes related to OS and CCS.
Comments on the Quality of English Language
Many sentences are too long, making them difficult to follow. Shorter sentences would improve readability.
Some phrases are repetitive or awkwardly worded, which affects clarity. For example, the sentence: “All this data underscores the potential of BAP1 expression as a tumor suppressor and make a support ground for our findings” is unclear.
Misuse of commas and unclear complex lists make some sections harder to read. Ensuring proper punctuation and restructuring long lists would improve clarity.
Response: We improved the readout through the manuscript
This study investigates the role of BRCA1-associated protein 1 (BAP1) expression in advanced and castration-resistant prostate cancer (PCa), specifically its association with disease prognosis and clinical outcomes. This study provides crucial insights into the molecular mechanisms of BAP1 in PCa and suggests it may be an important prognostic marker and therapeutic target for managing aggressive prostate cancer.
Nonetheless, there are some minor issues that require further discussion with the authors.
Comments:
- The clarity and logic of the research background needs to be improved
The clinical challenges of prostate cancer are somewhat touched upon in the introduction, but the role of BAP1 in prostate cancer is not described in depth. Although it is pointed out that BAP1 is a tumor suppressor and its role in other cancers has been partially clarified, there is a lack of detailed explanation regarding the specific research background of BAP1 in prostate cancer, related literature, and research gaps. Therefore, the clarity of the background is somewhat lacking, especially the specific biological functions of BAP1 and the current research status in prostate cancer have not been discussed in depth.
The article introduces the potential role of BAP1 and sets out the research objectives. However, the logic of how to identify research gaps from existing literature and why to focus on the relationship between BAP1 expression and prognosis in prostate cancer needs to be strengthened. For instance, it could further elaborate on why the role of BAP1 in prostate cancer has not been fully explored or what contradictions or deficiencies exist in the current research results.
Response: We revisited the manuscript sections and made additional changes as per reviewer suggestion
- The research methods are not elaborated in detail.·
The manuscript describes the analysis of 202 samples from PCa patients and verifies the association between BAP1 and the clinical prognosis of prostate cancer through methods such as gene set enrichment analysis (GSEA). The methods section is clear and suitable for the research purpose. Further elaboration on the statistical analysis methods (such as Cox regression analysis) can be provided, including how to control potential confounding factors (such as age, patient stage, etc.), and the reasons for choosing these methods. Regarding the differences between the TCGA PRAD dataset and the dataset of this study, it is suggested to further discuss the possible reasons and provide a detailed explanation of the data sources and specific analysis methods.
Response:
We have updated the method section to include these changes.
- The results section can be further improved.
The results section clearly demonstrates a significant relationship between low expression of BAP1 and poor clinical outcomes, such as a lower overall survival rate, while also showing the positive clinical prognosis of high BAP1 expression. Although the results are already clear, more specific data could be provided in the results section, such as the statistical significance level (p-value), hazard ratio (HR), etc., to enhance the operability and transparency of the results.
This manuscript conducts an in-depth exploration of the role of BAP1 in prostate cancer and its clinical significance. The research design is reasonable and the results are somewhat innovative. Nevertheless, it is suggested that the authors further elaborate on the background and methods sections of the study and provide more statistical details in the results to enhance the transparency and credibility of the research. Overall, this paper has high academic value and can be considered for publication after some revisions.
Comments on the Quality of English Language
The English could be improved to express the research more clearly.
Response: We improved the readout and writing throughout and edited the manuscript as needed.
Reviewer 2 Report
Comments and Suggestions for Authors
Section 2.1 is a bit unclear, which makes it harder to follow.
The use of two-tailed t-tests raises some concerns about statistical assumptions. It’s not clear whether the authors checked for normality before applying the test, which is important because violations of this assumption could affect the results. If the groups had unequal variances, Welch’s t-test would have been a better choice. Since multiple t-tests seem to have been used, it would be helpful to clarify whether any correction for multiple comparisons (such as a Bonferroni adjustment) was applied. If the data are not normally distributed, a nonparametric test like the Mann-Whitney U test might be more appropriate. Also, if other factors influence the comparisons, ANCOVA or regression models could provide a more reliable approach.
Figures 5C and 5D are too small to read—perhaps they could be rearranged for better visibility?
The discussion could be clearer in some areas. Many sentences are long and complicated, making it difficult to follow the key points. Breaking them into shorter sentences would improve readability. Some ideas are also repeated multiple times (for example, references to BAP1 as a prognostic factor and its connection to PTEN/ERG), which could be streamlined.
The authors mention that sample size is a limitation but don’t quantify how it impacts statistical power. Discussing effect sizes and confidence intervals would help strengthen their argument. They also note differences between their patient cohort and others but don’t fully explore potential confounding factors beyond treatment differences.
The discussion also touches on inconsistencies with a larger cohort study (~17,000 patients), but the explanation given (differences in treatment and cohort characteristics) feels a bit vague. It would be useful to consider whether other factors, such as patient selection criteria or analytical methods, could explain these discrepancies. Additionally, while the authors state that their findings align with Pan-Cancer data, they don’t clarify whether this applies specifically to prostate cancer or includes other tumor types, which could lead to confusion.
Comments on the Quality of English LanguageMany sentences are too long, making them difficult to follow. Shorter sentences would improve readability.
Some phrases are repetitive or awkwardly worded, which affects clarity. For example, the sentence: “All this data underscores the potential of BAP1 expression as a tumor suppressor and make a support ground for our findings” is unclear.
Misuse of commas and unclear complex lists make some sections harder to read. Ensuring proper punctuation and restructuring long lists would improve clarity.
Author Response

(The authors gave the same response as above.)

Reviewer 3 Report
Comments and Suggestions for Authors
This study investigates the role of BRCA1-associated protein 1 (BAP1) expression in advanced and castration-resistant prostate cancer (PCa), specifically its association with disease prognosis and clinical outcomes. This study provides crucial insights into the molecular mechanisms of BAP1 in PCa and suggests it may be an important prognostic marker and therapeutic target for managing aggressive prostate cancer.
Nonetheless, there are some minor issues that require further discussion with the authors.
Comments:
- The clarity and logic of the research background needs to be improved
The clinical challenges of prostate cancer are somewhat touched upon in the introduction, but the role of BAP1 in prostate cancer is not described in depth. Although it is pointed out that BAP1 is a tumor suppressor and its role in other cancers has been partially clarified, there is a lack of detailed explanation regarding the specific research background of BAP1 in prostate cancer, related literature, and research gaps. Therefore, the clarity of the background is somewhat lacking, especially the specific biological functions of BAP1 and the current research status in prostate cancer have not been discussed in depth.
The article introduces the potential role of BAP1 and sets out the research objectives. However, the logic of how to identify research gaps from existing literature and why to focus on the relationship between BAP1 expression and prognosis in prostate cancer needs to be strengthened. For instance, it could further elaborate on why the role of BAP1 in prostate cancer has not been fully explored or what contradictions or deficiencies exist in the current research results.
- The research methods are not elaborated in detail.ï‚·
The manuscript describes the analysis of 202 samples from PCa patients and verifies the association between BAP1 and the clinical prognosis of prostate cancer through methods such as gene set enrichment analysis (GSEA). The methods section is clear and suitable for the research purpose. Further elaboration on the statistical analysis methods (such as Cox regression analysis) can be provided, including how to control potential confounding factors (such as age, patient stage, etc.), and the reasons for choosing these methods. Regarding the differences between the TCGA PRAD dataset and the dataset of this study, it is suggested to further discuss the possible reasons and provide a detailed explanation of the data sources and specific analysis methods.
- The results section can be further improved.
The results section clearly demonstrates a significant relationship between low expression of BAP1 and poor clinical outcomes, such as a lower overall survival rate, while also showing the positive clinical prognosis of high BAP1 expression. Although the results are already clear, more specific data could be provided in the results section, such as the statistical significance level (p-value), hazard ratio (HR), etc., to enhance the operability and transparency of the results.
This manuscript conducts an in-depth exploration of the role of BAP1 in prostate cancer and its clinical significance. The research design is reasonable and the results are somewhat innovative. Nevertheless, it is suggested that the authors further elaborate on the background and methods sections of the study and provide more statistical details in the results to enhance the transparency and credibility of the research. Overall, this paper has high academic value and can be considered for publication after some revisions.
Comments on the Quality of English LanguageThe English could be improved to more clearly express the research.
Author Response

(The authors gave the same response as above.)

Round 2
Reviewer 3 Report
Comments and Suggestions for Authors
Comparing the first review opinions with the revised manuscript, it can be seen that the authors have revised and improved in some aspects, but there are still some key problems that have not been completely solved. Comparing the first review opinions with the revised manuscript, it can be seen that the authors have revised and improved in some aspects, but there are still some key problems that have not been completely solved.
Minor Comments:
- The research background is insufficiently improved.
Some background information on the role of BAP1 in cancer was added in the revised manuscript. However, the literature review on the specific role of BAP1 in prostate cancer remains insufficient. The article still lacks a systematic review of the current research status and controversies regarding BAP1 in prostate cancer, such as the differences in BAP1 expression levels among different patient groups and the contradictory conclusions from different experimental studies. It is suggested that the authors add a more in-depth literature review in the introduction section, for example: Have existing studies explored the specific function of BAP1 in prostate cancer? Has the role of BAP1 in castration-resistant prostate cancer (CRPCA) been conclusively determined? What are the new findings and breakthroughs of this study compared to existing research?
- The discussion section is still relatively weak.
The revised manuscript still mainly emphasizes the role of BAP1 as a tumor suppressor gene and does not fully discuss the contrary evidence. There is still a lack of mechanism-based exploration as to why low expression of BAP1 is associated with a poor prognosis.
- The results section can be further improved.
It is necessary to explain more clearly in the results section the differences in the TCGA PRAD data, for example: Could there be post-translational modifications or degradation mechanisms of the BAP1 protein that lead to an inconsistency between the protein expression level and the mRNA level?
Comments on the Quality of English Language
This paper still has room for improvement in terms of language expression and discussion.
Author Response
Response letter
Minor Comments:
- The research background is insufficiently improved.
Some background information on the role of BAP1 in cancer was added in the revised manuscript. However, the literature review on the specific role of BAP1 in prostate cancer remains insufficient. The article still lacks a systematic review of the current research status and controversies regarding BAP1 in prostate cancer, such as the differences in BAP1 expression levels among different patient groups and the contradictory conclusions from different experimental studies. It is suggested that the authors add a more in-depth literature review in the introduction section, for example: Have existing studies explored the specific function of BAP1 in prostate cancer? Has the role of BAP1 in castration-resistant prostate cancer (CRPCA) been conclusively determined? What are the new findings and breakthroughs of this study compared to existing research?
Response# Thank you for giving detailed feedback. We agree with your suggestion and we added to the manuscript the details required.
- The discussion section is still relatively weak.
The revised manuscript still mainly emphasizes the role of BAP1 as a tumor suppressor gene and does not fully discuss the contrary evidence. There is still a lack of mechanism-based exploration as to why low expression of BAP1 is associated with a poor prognosis.
Response# Thank you suggestions. We reedited the discussion with more details as per your feedback.
- The results section can be further improved.
It is necessary to explain more clearly in the results section the differences in the TCGA PRAD data, for example: Could there be post-translational modifications or degradation mechanisms of the BAP1 protein that lead to an inconsistency between the protein expression level and the mRNA level?
Response# We looked further into the details regarding the TCGA data and we could not identify any additional data that can be added regarding the post translational modifications or proteins degradation.